# Localization, inspection, and reasoning (LIRA) module for autonomous workflows in self-driving laboratories
Zhengxue Zhou [1,2], Satheeshkumar Veeramani [1], Francisco Munguia-Galeano [1], Hatem Fakhruldeen[1,2] & Andrew I. Cooper [1,2] ✉

Self-driving labs (SDLs) combine robotic automation with artificial intelligence (AI) to allow autonomous, high-throughput experimentation. However, robot manipulation in most SDL workflows operates in an open-loop manner, lacking real-time error detection and error correction. This can reduce reliability and overall efficiency. Here, we introduce LIRA (Localization, Inspection, and Reasoning), which is an edge computing module that enhances robotic decision-making through vision-language models (VLMs). LIRA enables precise localization, automated error inspection, and reasoning, thus allowing robots to adapt dynamically to variations from the expected workflow state. Integrated within a client-server framework, LIRA supports remote vision inspection and seamless multi-platform communication, improving workflow flexibility. Through extensive testing, LIRA achieves high localization accuracy, a tenfold reduction in localization time, and real-time inspection across diverse tasks, increasing the efficiency and robustness of autonomous workflows considerably. As an open-source solution, LIRA facilitates AI-driven automation in SDLs, advancing autonomous, intelligent, and resilient laboratory environments. Longer term, this will accelerate scientific discoveries through more seamless human-machine collaboration.

Self-driving laboratories (SDLs) have evolved rapidly over the last 5 years, in part through advances in automation. This has been driven by the growing demand for high-throughput experimentation in chemistry and materials science[1]. There has been a marked shift beyond more traditional self-contained lab automation systems toward more modular, adaptable, and distributed solutions, enabled by collaborative robots. Examples of this more modular approach include sample preparation[2] and synthesis workflows that require coordination of multiple subsystems, including liquid chromatography-mass spectrometry (LCMS) and nuclear magnetic resonance (NMR) spectroscopy[3]. However, robotic manipulation in these autonomous workflows operates in an open-loop manner, which assumes flawless execution without accounting for potential failures. This is an unrealistic assumption because robot manipulation errors can be relatively common in practice. Moreover, human errors, such as loading laboratory consumables in the wrong areas, further increase the chance of workflow failure. In traditional, non-automated laboratories, human chemists rely on real-time visual inspection to detect, reason and correct errors to ensure precision when interacting with scientific equipment. Likewise, to exploit their full potential, SDLs will require closed-loop manipulation modules to

monitor, validate, and dynamically recover from errors in robotic tasks, thus enabling seamless integration with analytical instruments and ensuring safe and precise execution[4]. For example, workflows involving PXRD plate insertion and LCMS/NMR vial analysis require high-precision placement to avoid failed measurements or instrument damage, making robust error inspection critical in real chemical settings.

Current implementations of robotic manipulators in SDLs fall into two primary architectures: fixed-base robots and mobile-base robots. Fixed-base robots excel in precise (albeit not infallible) positioning within confined workspaces but lack the mobility required to interact with spatially distributed laboratory instruments. To address this limitation, mobile-base systems have been introduced, integrating mobility either through robots equipped with linear translational tracks[5,6] or wheeled platforms[3,7–9], thereby expanding their accessible workspace and enabling interactions across larger laboratory environments. However, the inherent navigation inaccuracies of mobile bases can compromise precision-critical manipulation tasks in SDLs. To mitigate this, tactile-based localization methods, such as cube-mounted location systems, can achieve high accuracy[3,7,8] but require static infrastructure, limiting their flexibility in dynamic environments. By

[1]Department of Chemistry and Materials Innovation Factory, University of Liverpool, Liverpool, United Kingdom. [2]The Leverhulme Research Centre for Functional Materials Design, University of Liverpool, Liverpool, United Kingdom. ✉e-mail: aicooper@liverpool.ac.uk

contrast, vision-based methods (e.g., fiducial markers, LIDAR, or learning-based keypoint detection) offer adaptability for dynamic workflows while maintaining precision[9–17].

Recent advances in integrating machine vision with robotics have enabled more perceptive and adaptable systems. Methods such as keypoint detection[13,14], semantic segmentation, and 6D pose estimation enhance scene understanding and manipulation. End-to-end approaches such as visual servoing and closed-loop feedback improve task precision in applications such as bin picking and peg-in-hole operations[14,18]. However, these promising strategies remain underused in SDLs, where vision is so far typically limited to navigation or static detection. Task-level visual reasoning-such as verifying sample vial placement or instrument door closure-is rarely implemented. Existing models for SDL robots often lack semantic understanding to interpret and correct manipulation failures. LIRA addresses this gap with a unified framework that enables real-time visual inspection and reasoning via vision-language models.

However, even with vision, minor localization errors-which are common in vision-based systems-can lead to failures in high-precision tasks, such as peg-in-hole operations[14,19] and, by extension, SDLs. Localization alone assumes ideal conditions and does not account for execution uncertainties, resulting in an open-loop manipulation where errors accumulate without correction. Incorporating additional vision feedback on the environment has the potential to greatly enhance robotic manipulation reliability in SDLs, improving adaptability and error resilience[20]. Building on this paradigm, the latest progress in machine learning has enhanced vision-based methods, enabling automated visual detection for analyzing material states (e.g., phase boundaries[21,22], crystallization[23]) and process anomalies (e.g., bubble formation[24], liquid level changes[23,25,26]). Nonetheless, existing deep learning (DL) approaches in SDLs remain narrowly focused: they primarily address object detection (e.g., glassware[16,27,28], syringe needles[9]) and the like, rather than tackling broader workflow challenges such as error inspection and manipulation validation. For instance, in SDLs, vials and racks are routinely transferred between stations (e.g., liquid dispensers to LCMS/NMR systems[3]), and these pick-and-place insertions rely on assumptions of flawless execution, when in fact they are a common failure point. Recent work has proposed using DL to verify vial placement[29], yet such methods remain limited to specific tasks, requiring extensive task-specific training data and lacking generalizability to novel scenarios. This lack of generalization makes it difficult for DL-based detection methods to adapt to general-purpose error inspection in SDLs, since they struggle both with dynamic errors and also lack semantic understanding, limiting their ability to perform holistic reasoning across workflows.

Recent advances in large language models (LLMs) show promise for addressing inspection challenges in SDLs, leveraging their natural language understanding and perceptual capabilities to diagnose and recover from task-level errors such as workflow deviations[30–34]. VLMs, which integrate LLMs with visual perception, further enhance robotic

proprioception[18,35–37], yet significant challenges remain in detecting and resolving manipulation-level failures (e.g., misalignments, dropped objects, etc.), which are critical for SDLs operating under stringent safety protocols[38]. Unlike human chemists, who can instantly recognize and interpret errors such as an offset reaction vial, SDL robots lack intrinsic error awareness and require autonomous systems for real-time failure detection and correction. Bridging this gap demands frameworks that tightly integrate VLMs into SDL workflows, enabling seamless coordination of perception, reasoning, and manipulation for robust error inspection. VLMs hold promise for empowering mobile manipulators with real-time inspection capabilities to enhance workflow robustness, but their application to SDLs remains underexplored. The computational demands of real-time VLM inference strain the limited resources of edge computing modules deployed on mobile platforms.

LIRA is positioned to become a critical reasoning module within a fully autonomous self-driving lab. While it currently functions as a modular perception and decision-making unit, its structured outputs-such as error classifications and recovery strategies-can serve as inputs to task schedulers and robotic control systems. This opens the door to tighter integration with high-level planners and workflow engines, enabling closed-loop execution without human intervention. In future iterations, LIRA can act as a bridge between perception and action, automatically triggering retries, task reordering, or experiment replanning based on real-time visual feedback. Its modularity and low-latency operation make it well-suited for seamless integration into fully autonomous SDLs.

Here, we report LIRA, an edge computing module that enhances mobile manipulators with real-time perception and decision-making, enabling closed-loop robotic workflows in SDLs. As illustrated in Fig. 1b, LIRA integrates vision-based localization, inspection, and reasoning into a unified framework, allowing robots to detect and correct manipulation errors autonomously, improving workflow robustness. By leveraging an edge computation device, LIRA processes visual inputs dynamically, reducing reliance on predefined task sequences and increasing adaptability in dynamic lab environments. The integrated VLM is fine-tuned on an image dataset of a chemistry lab, which is a semi-structured environment, allowing it to generalize to similar settings and support decision-making in robotic tasks. Compared to tactile-based localization methods, LIRA enhances localization flexibility by using a vision-based approach with a calibration board, eliminating the need for direct force-based interaction. This simplifies system deployment while maintaining high localization accuracy. To validate its effectiveness, we tested LIRA with tasks in two autonomous workflows[3,8] as shown in Fig. 1a, c. With minimal setup adjustments, LIRA improves execution efficiency, reducing by 34% manipulation time in the solid-state workflow illustrated in Fig. 1a and enabling remote inspection for a robot in the synthesis workflow as shown in Fig. 1b. These enhancements both optimize

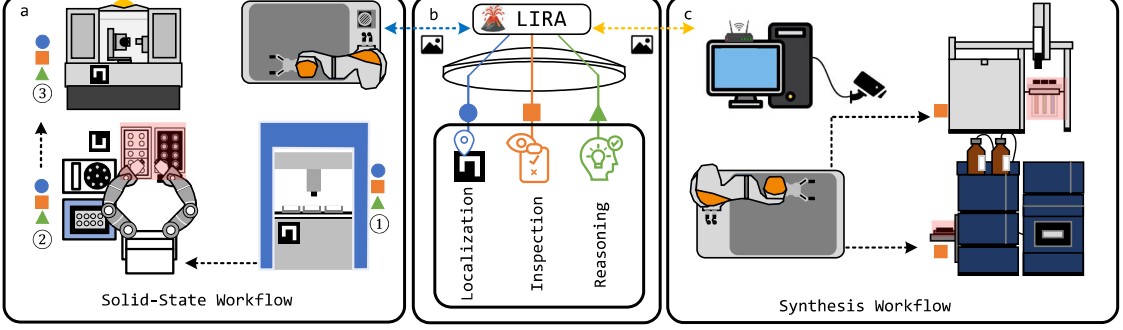

**Fig. 1 | Overview of LIRA's functionality and integration into SDL workflows. a** LIRA-enabled solid-state materials workflow, enhancing robotic manipulation through real-time localization, inspection, and reasoning. **b** Core functionalities of LIRA, including localization (blue circle), inspection (orange square), and reasoning (green triangle). **c** LIRA-enabled synthesis workflow, demonstrating remote access and integration with external systems.

**Fig. 2 | Schematic design of LIRA. a** The example of ArUco client and inspection client. **b** The architecture of LIRA comprises two main components: the Aruco detection node and the VLM inspection node. The Aruco detection node is used to detect the poses of fiducial markers to localize the robot's end-effector pose with respect to workstations by updating the robot's frames. The VLM node provides the function for inspection and reasoning the manipulation error.

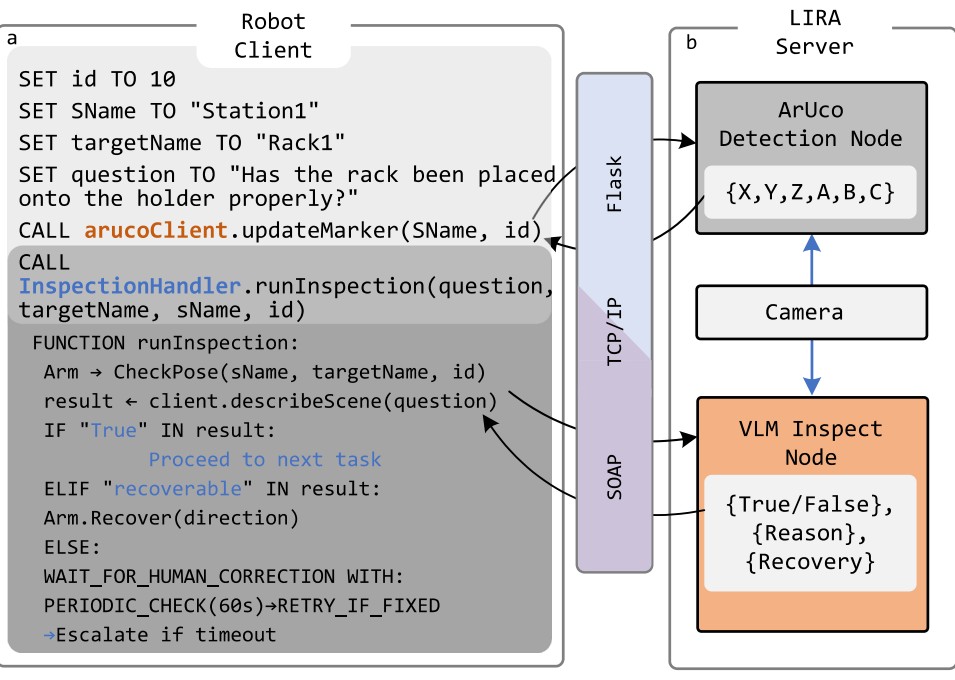

robotic operation and also demonstrate LIRA's scalability across SDL platforms. Furthermore, LIRA achieves an error inspection success rate of 97.9%, confirming its reliability in detecting and addressing task-level failures. LIRA represents a step toward more autonomous and self-correcting experimental workflows by enabling real-time inspection and reasoning, bridging the gap between robotics, computer vision, and AI-driven decision-making in SDLs.

## Results

### Architecture of LIRA

The architecture of LIRA is illustrated in Fig. 2. LIRA comprises three layers: the robot client interface, the communication layer, and the server layer. The client interface is implemented as a command-line interface, enabling users to import client classes and integrate localization, inspection, and reasoning functions directly into existing workflow scripts. The communication layer detailed in Method 4.3, operating over a local network using both Simple Object Access Protocol (SOAP) and Flask, manages data exchange-such as text requests paired with camera images-between system components. The server layer, running on an edge device, delivers the computational power required for real-time image processing and executing the fine-tuned VLM which is illustrated in Fig. 2b.

As shown in the pseudocode example in Fig. 2a, workflows initialize parameters (e.g., station name, target name, fiducial marker id), perform vision-based localization, and inspection for closed-loop error recovery. After the mobile robot navigates to a station, the arm moves to a predefined position for visual calibration. The client sends an ArUco pose request via the network with the function of `arucoClient.updateMarker`, and a $6 \times 1$ vector representing the marker pose is returned to update the manipulation-related frames at that station.

Subsequently, the robotic arm executes its manipulation task. At critical points where errors are likely to occur, the inspection client is invoked. An inspection request, formulated as an input prompt, is transmitted while the camera captures an image. As outlined in the pseudocode in Fig. 2a, this information is processed by LIRA on the server, which returns the inspection and reasoning results to the client. The robot then applies the `InspectionHandler` to interpret the response and determine the appropriate next action. Further details on the `InspectionHandler` are provided in Supplementary Section S.7.

### Localization precision evaluation

Using fiducial markers to assist robot localization can encounter challenges such as motion blur, non-uniform lighting, and reduced visibility[39]. To address these issues, we employ a combination of Gaussian and mean filters to refine the visual information, which is detailed in Method 4.4. To assess localization precision, we evaluated the robot's end-effector accuracy after calibrating with fiducial markers of different sizes ($20 \times 20$ mm to $75 \times 75$ mm ArUco markers) and compared vision-based and tactile-based calibration methods.

For accuracy assessment, we defined a reference position 300 mm away from the calibration marker position, measured along the robot base coordinate system, as shown in Fig. 3a. This distance was determined using the robot's internal position measurement capabilities, leveraging the intrinsic positional precision of the KUKA robotic arm, which has a documented pose repeatability of ±0.15 mm (ISO 9283). The ground truth for the reference position was established through tactile calibration using a cube-mounted locator, a method extensively validated in our laboratory and known to achieve sub-millimeter accuracy[7,8]. After performing tactile-based calibration, the robot's end-effector pose at the reference position was recorded and used as the benchmark for evaluating the vision-based calibration accuracy.

After each calibration, the robotic arm was commanded to move to this target, and the recorded end-effector pose was compared against the ground truth. This process was repeated 100 times to ensure statistical reliability, allowing for a comprehensive evaluation of calibration accuracy across different conditions. Positional and rotational deviations were computed as the translational and rotational errors from the ground truth position, respectively.

Figure 3b shows the box plots comparing the calibration accuracy across different methods for both translation and rotation errors. The 20 mm ArUco marker before filtering exhibits the highest error and greatest variance in both translation and rotation accuracy, with a mean translation error of 2.42 mm and a mean rotation error of 3.46 degrees. After applying Gaussian and mean filtering, the errors reduce significantly to 1.51 mm and 1.54 degrees, respectively. The 75 mm ArUco marker further improves stability, achieving a mean error of 1.02 mm in translation and 1.01 degrees in rotation. The cube-based calibration method demonstrates the highest precision, with minimal variance and a mean error of 0.92 mm in translation and 0.90 degrees in rotation. These results confirm that filtering

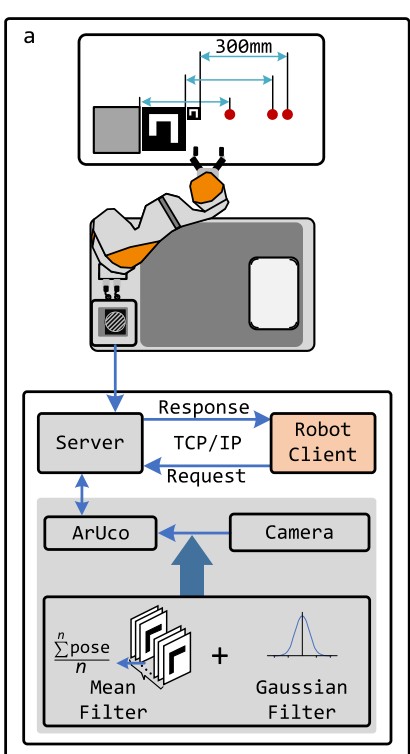

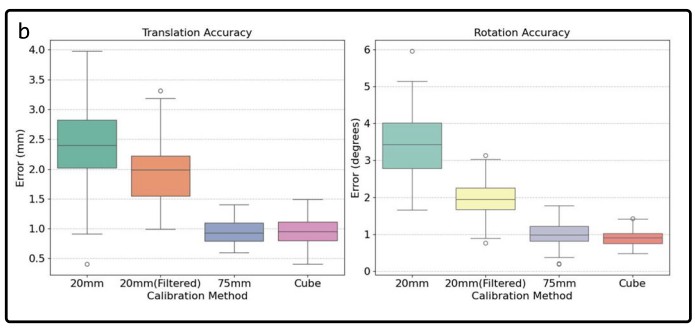

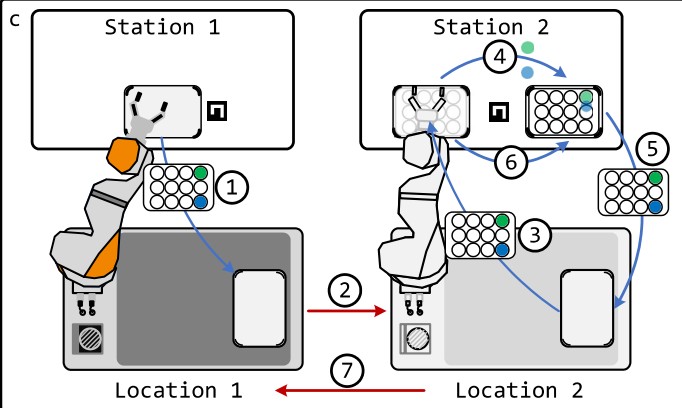

**Fig. 3 | Evaluation of vision-based localization and stability testing. a** The localization precision evaluation and filtering techniques applied to fiducial marker detection. **b** Box plots comparing calibration accuracy across different methods for both translation and rotation errors. **c** Stability testing procedure for evaluating communication reliability, localization accuracy, and manipulation precision.

techniques and marker size enhance calibration accuracy to a level that is comparable with tactile cube location, making the vision-based approach a viable alternative to tactile localization while enhancing adaptability in SDL environments.

In our implementation, the calibration board is affixed to stationary laboratory instruments or robotic workstations. This attachment implicitly couples the spatial frames defined by the calibration process with the physical hardware, such that these frames are consistently preserved and effectively co-located with the instrument. Consequently, when laboratory layouts are reconfigured, relocating the calibration board together with its associated machine enables the re-establishment of spatial references with minimal recalibration effort. This benchmark-based approach provides a pragmatic and low-overhead solution for current SDL infrastructures, balancing localization accuracy with practical deployment constraints. In future iterations, markerless calibration strategies-such as those leveraging learned keypoints or contextual visual features-may further reduce the need for human involvement while enhancing adaptability in dynamic laboratory environments.

## Operational stability evaluation

To evaluate the robustness of the LIRA module in real-world conditions, we conducted an operational stability evaluation focusing on communication reliability, localization accuracy, and manipulation precision. Specifically, we aimed to validate stable client-server communication and investigate the impact of mobile base navigation errors on vision-based localization performance, ensuring the system's adaptability to dynamic environments.

The evaluation was conducted through a repeated vial transfer workflow, designed to simulate a prolonged autonomous experiment. Figure 3c illustrates the testing procedure. In each loop, the robot first updated its localization using ArUco markers (20 mm × 20 mm) and then executed a series of precise manipulations. At Station 1 (step ①), the robot picked up a rack containing two vials and placed it onto the mobile base's rack holder. The robot then navigated to Station 2 (step ②) and positioned the rack onto

Rackholder 1 (step ③). The vials were then transferred into an empty rack (step ④), after which the robot returned the filled rack to the mobile base (step ⑤). Finally, the previously emptied rack was placed onto Rackholder 2 (step ⑥), and the robot returned to its initial position (step ⑦) to begin the next loop. Throughout the experiment, navigation errors introduced by the mobile base tested the system's ability to adapt its localization processes. The use of small ArUco markers added complexity by challenging the system's ability to localize and execute precise manipulations accurately under real-world conditions.

Over 100 consecutive workflow loops, each lasting ~5 min, the system operated continuously for 8 h without failure. Across 600 pickup and insertion manipulations, the LIRA module achieved a 100% success rate, demonstrating its high operational stability, resilience to navigation errors, and suitability for long-term, high-precision workflows in SDL environments. A demonstration video is provided in Supplementary S1 (The authors affirm that informed consent for publication of the images in the Supplementary Video 1 was obtained from the identifiable individuals.).

## VLM for inspection and reasoning

Figure 4a illustrates how robots were used to collect an image dataset for fine-tuning the VLM for error inspection and recovery reasoning. The dataset includes images of properly placed and misaligned sample racks used in our previous work[3,8] (i.e., 8-hole racks, PXRD plates, NMR racks, and LCMS racks) captured at a resolution of 640 × 480 pixels. Specifically, images were collected using a camera mounted on the robot's end effector, providing consistent viewpoints directly above the racks. Misalignments were systematically introduced by manually offsetting the racks in distinct translational directions (Left, Right, Forward, Backward) as well as rotational misalignments (LRFB+R). The coordinate system is defined from the camera's perspective, with L (Left), R (Right), F (Forward), and B (Backward) indicating offset directions. Misalignment cases were manually introduced by placing racks with different offsets, resulting in a dataset of

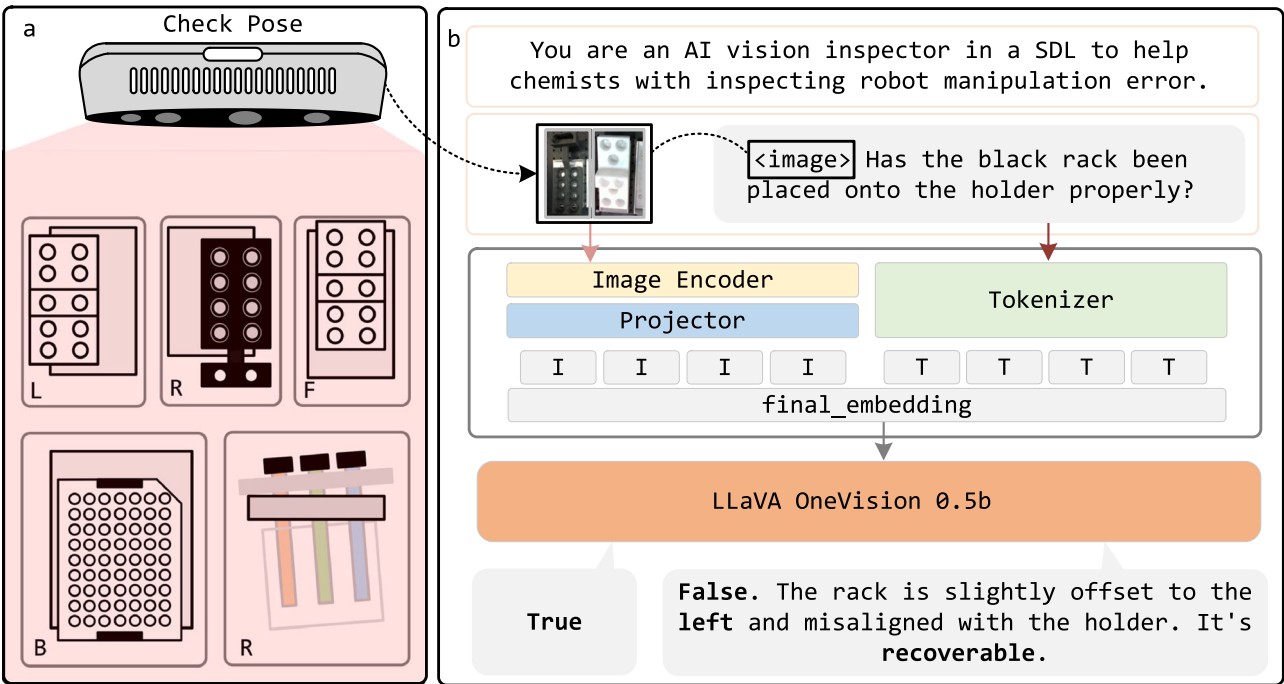

**Fig. 4 | Overview of VLM fine-tuning for inspection and reasoning. a** Data collection process for fine-tuning the VLM using robotic image acquisition. **b** Structured prompt format used for inspection, reasoning, and recovery tasks.

540 images for robotic inspection and reasoning tasks. Examples of the dataset used for fine-tuning are provided in Supplementary S.4.

A structured data annotation process was implemented to equip LIRA with both inspection and reasoning capabilities. Each image was manually labeled by an experienced operator to indicate: (1) whether the object placement was correct or incorrect, (2) the direction of the misalignment (Left, Right, Forward, Backward, or Rotated), and (3) whether the error was considered recoverable. These labels were paired with carefully designed prompts to supervise fine-tuning. Task-specific prompts structured to guide model behavior were used to fine-tune the VLM inspector, ensuring accurate error detection, reasoning, and recovery for manipulation tasks. The prompt format remains consistent across all tasks, as illustrated in Fig. 4b. For each inspection task, a standardized human prompt is used. For example, to inspect the 8-hole rack for Chemspeed vials, the following structured prompt is used:

```
Has the white 8-hole rack been placed onto the holder
properly?
```

LIRA responds with either a True or False statement. If the response is False, LIRA further reasons the misalignment and evaluates whether it is recoverable.

Example response:

```
False. The white rack is slightly offset to the back and
misaligned with the holder. It is recoverable.
False. The white rack is slightly rotated and mis
aligned with the holder. It is unrecoverable.
```

When the response is False, the model first determines the type of misalignment by identifying the offset direction (left, right, back, forward, rotated) relative to the camera frame. If the error is purely translational, it is considered recoverable, and the robot executes a predefined correction strategy corresponding to the identified direction of misalignment. However, if the error involves a rotational misalignment, it is deemed unrecoverable, requiring human intervention. The response is handled inside the

`InspectionHandler`, which is detailed in Supplementary S.7. The prompt dictionary is provided in Supplementary S.4.

To benchmark the performance of our VLM against a conventional convolutional approach, we fine-tuned a YOLOv11[40] classification model on the same dataset. The YOLOv11 model achieved a classification accuracy of 86.3%, indicating competent but limited performance in this complex inspection task. Training and classification details are provided in Supplementary S.6.

In contrast, we fine-tuned LLaVA-OneVision 0.5B[41] using Low-Rank Adaptation of Large Language Models (LoRA)[42]. The final model demonstrated strong convergence with a training loss of 0.0123 and an evaluation loss of 0.0424, ensuring reliable inspection and reasoning performance. To validate the effectiveness of fine-tuning, we also evaluated the original pretrained model (LLaVA-OneVision 0.5B) in a zero-shot setting using the same test dataset. The zero-shot model achieved an overall success rate of 42.9%, correctly identifying placement in 77.7% of cases, but showing limited performance in direction classification (25.5%) and recoverability prediction (25.5%). In contrast, the fine-tuned model achieved a significantly improved overall accuracy of 97.9%, with respective accuracies of 97.9% for placement, 96.8% for direction, and 98.9% for recoverability. This highlights the benefit of domain adaptation for task-specific reasoning in SDL environments. Detailed results and per-sample breakdowns are provided in Supplementary S.5.

## Inspection and reasoning workflows

The experiment for evaluating LIRA is conducted as shown in Fig. 5, which illustrates its integration across two representative SDL workflows: a solid-state workflow[8] and an autonomous synthesis workflow[3]. In the solid-state workflow, the PXRD plate must be precisely inserted into the diffractometer for accurate material characterization. Misalignment can lead to failed measurements or the need for manual correction, interrupting the automation pipeline. LIRA enables visual verification of plate placement before and after insertion, reducing these risks. In the synthesis workflow, racks containing product vials are transferred to LCMS and NMR for purity analysis. If these racks are mispositioned, autosampler needles may miss target vials or damage themselves. LIRA inspects the setup before

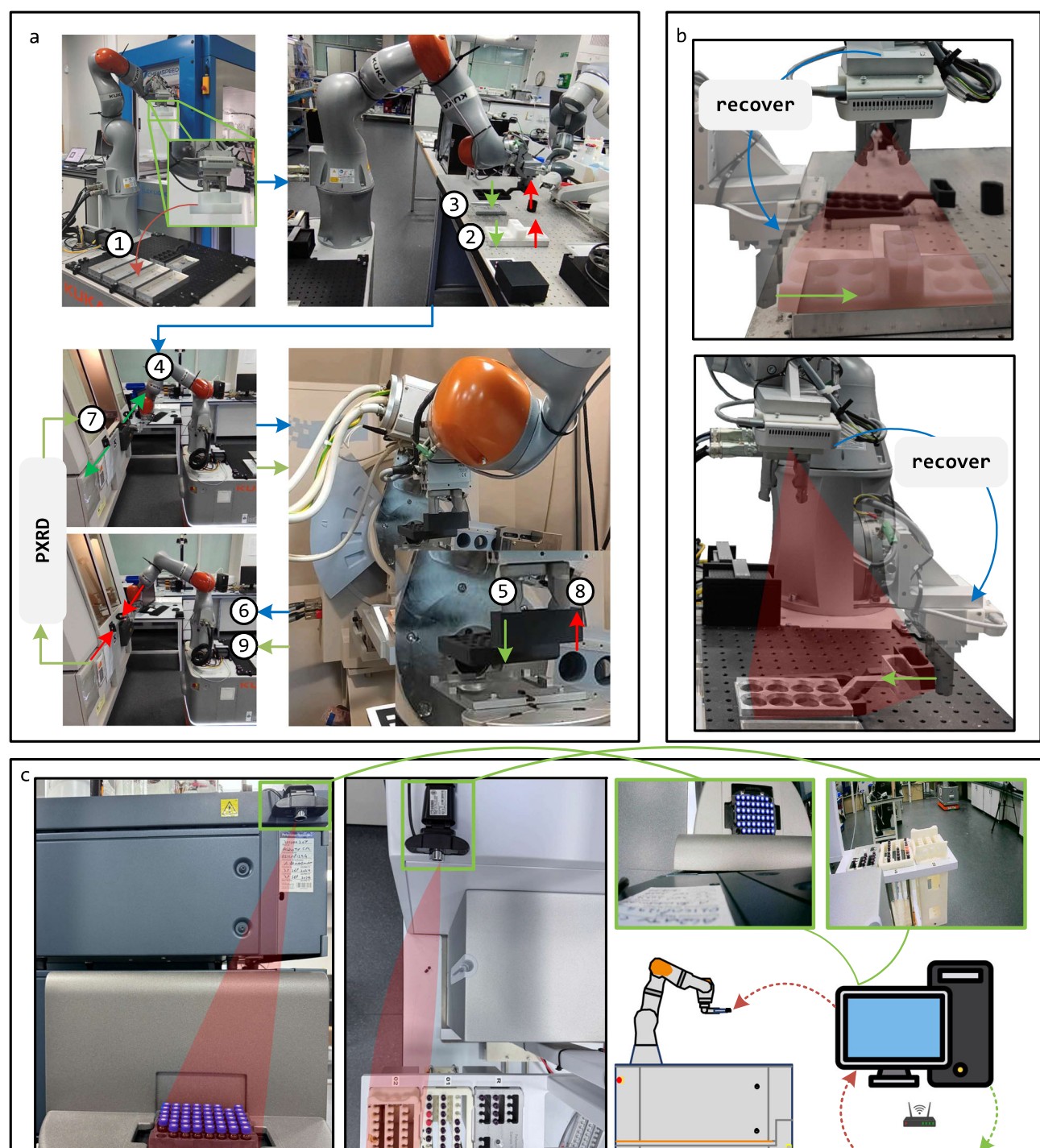

**Fig. 5 | Enhanced SDL workflows with LIRA for localization, inspection, and reasoning. a** Solid-state workflow incorporating LIRA for visual localization and closed-loop placement verification. **b** Example of LIRA performing inspection and reasoning after rack and plate placement, enabling error detection and correction. **c** Example of LIRA performing inspection and reasoning for LCMS and NMR racks in the synthesis workflow, with remote processing.

measurements begin, preventing these issues and ensuring accurate data collection.

The robotic platform used is a KUKA KMR iiwa mobile manipulator, which performs object handling and transfers between stations. Figure 5a provides an overview of the full solid-state workflow pipeline, where LIRA is primarily used for vision-based localization using fiducial markers. Figure 5b zooms into the robot's closed-loop manipulation process, showing how a camera mounted on the robot's end effector captures images post-

placement for LIRA to perform inspection and reasoning. Figure 5c demonstrates LIRA's application in the synthesis workflow, where cameras fixed above LCMS and NMR stations stream images wirelessly to the Jetson AGX Orin for remote error detection and reasoning.

Across all workflows, different camera configurations are employed depending on context: the RealSense D435i camera mounted on the robot's end effector is used for precise post-placement inspection and localization, while fixed webcams are deployed on laboratory instruments for remote

**Table 1 | Inspection and reasoning tasks across two different workflows**

| Workflow | Stations | Inspection targets | Reasoning | Workflow steps |
|---|---|---|---|---|
| Solid-state workflow | Base Station | Rack, Plate | Placing offset, Error recoverability | ①②③⑧ |
| | Preparation Station | Rack, Plate | Placing offset, Error recoverability | ②③ |
| | PXRD Station | Door, Plate | Door State, Placing offset, Error recoverability | ④⑤⑥⑦⑨ |
| | Chemspeed synthesizer | Rack | Placing offset, Error recoverability | – |
| Synthesis workflow | LCMS Station | LCMS Rack | Rack Placing Offset | – |
| | NMR Station | NMR Rack | Rack Placing Offset | – |

inspection scenarios. Regardless of source, all image data is processed by the same LIRA module, allowing consistent reasoning logic across embodiments and workflows.

The solid-state workflow involves multiple stations such as Chemspeed synthesizer, PXRD, and sample preparation modules. Figure 5a outlines the solid-state workflow enhanced by LIRA. The sequence begins with the robot picking up an 8-hole rack from the Chemspeed synthesizer and placing it onto the base holder (step ①). The robot then navigates to the preparation station, where it sequentially loads the 8-hole rack and PXRD plate onto the workstation (steps ② and ③) before waiting for the grinding step (detailed in the ref. 3). After grinding, the robot unloads the rack and plate from the station and returns them to the base rack holders. Next, the robot navigates to the PXRD station and opens the instrument door (step ④). It then loads the PXRD plate into the instrument (step ⑤), using a calibration board inside the PXRD machine to ensure accurate positioning, before closing the door (step ⑥). Once the PXRD data acquisition is complete, the robot unloads the plate (step ⑧) following the door operation (steps ⑦ and ⑨). Overall, this one of the most complex workflows that we have built in our laboratories so far, and it has multiple potential failure points.

All manipulation tasks, except for placing the rack and plate on the mobile robot base, rely on the vision-based localization method with fiducial markers attached to each station. These operations are further verified by LIRA's inspection and reasoning capabilities, ensuring precise execution throughout the workflow. A detailed breakdown of the inspection and reasoning tasks across the two workflows is provided in Table 1. As shown in the table, in the solid-state workflow, each placement manipulation operates in a closed-loop manner, with verification handled by LIRA's inspection and reasoning functions. A single workflow iteration consists of 11 placement manipulations, each of which is verified in a closed-loop manner using LIRA's inspection and reasoning capabilities. After each object (e.g., a rack or PXRD plate) is placed at its target location, the robot moves its end effector to a predefined inspection pose. An image is then captured and sent to LIRA for analysis. LIRA evaluates the placement by responding to a standardized prompt with a structured output indicating, as illustrated in Fig. 4b: (1) whether the placement is correct, (2) the direction of any misalignment, and (3) whether the error is recoverable. If LIRA returns "False" with a recoverable offset, the InspectionHandler parses the reasoning and commands the robot to execute a predefined corrective motion (e.g., a small push in the identified direction). The inspection is repeated after each correction, and this loop continues iteratively until LIRA returns "True," confirming that the object is correctly placed. If the error is deemed unrecoverable (e.g., a rotational misalignment), the workflow is paused, and human intervention is requested. This closed-loop verification mechanism ensures robust and autonomous correction of common placement errors, enhancing the reliability of each manipulation task in the workflow. For example, in Fig. 5b, the robot places a rack and a plate onto the base and moves to the inspection pose. LIRA then provides inspection and reasoning results, which are processed by the InspectionHandler, which is detailed in Supplementary S.7 to determine whether the workflow should proceed or trigger an error recovery process.

In the synthesis workflow, local cameras mounted on the LCMS and NMR instruments capture images for inspection as shown in Fig. 5c. The cameras are connected to a local PC, which runs the inspection client and

sends the images to LIRA over a local wireless network. LIRA then processes the images and returns inspection and reasoning results. Remote users only need to implement a python client to interpret these results for decision-making and robotic guidance. The python client is detailed in Method 4.3. In this setup, error recoverability was not considered, as both LCMS and NMR racks are placed in constrained spaces, making robotic intervention impractical.

For both the solid-state and synthesis workflows, we conducted 200 manual tests, moving the racks and plates to introduce placement errors and then inspecting and reasoning the misalignments with LIRA. The system achieved a 97.5% accuracy rate, with only 5 failures-2 related to PXRD plate placement and 3 involving the NMR rack.

To evaluate the impact of LIRA on robotic arm manipulation, we compared the enhanced solid-state workflow with the previous version[8] in terms of picking, placing, and localization time. Before integrating LIRA, the robot performed seven tactile-based calibrations, each taking ~53 s. With LIRA, the workflow now includes 11 visual-based calibrations, each completed in 5 s. The high efficiency of visual localization allows calibration for each placement, enabling the use of multiple fiducial markers per station to maintain manipulation accuracy, which is otherwise sensitive to distance variations as shown in Fig. 3. The robot arm operates at the same speed in both workflows, with the current LIRA-enhanced workflow requiring 873 s per loop, including 11 inspection and reasoning cycles. In contrast, the previous workflow required 1367 s for arm manipulation, with 27% of the time spent on tactile-based localization. Overall, LIRA reduces the total arm manipulation time by 36.1%, while also providing greater flexibility in marker placement and a more robust closed-loop manipulation strategy. Importantly, all testing and benchmarking were performed in real-world robotic workflows, with the LIRA module invoked directly from robot-side programs via the inspection handler interface as shown in the Supplementary S.7. These results reflect integrated, continuous system operation rather than isolated module evaluation.

During testing, no placement offset was observed, allowing us to validate LIRA's inspection and reasoning capabilities in correcting recoverable placement errors, as demonstrated in Supplementary S.3. This comprehensive testing effectively showcases the robustness and adaptability of LIRA in SDL environments.

## Discussion

LIRA presents a robust solution for enhancing autonomous robotic workflows in SDLs, addressing and recovering manipulation failure through VLM-driven inspection and reasoning. To overcome domain-specific limitations of VLMs, we fine-tuned the model with robot manipulation failure dataset in the semi-structured environment of a laboratory, ensuring improved reasoning capabilities tailored to SDL workflows.

LIRA was deployed onto an edge-computation module as a proof-of-concept, equipping a KUKA KMR iiwa mobile manipulator with advanced vision capabilities. The system was evaluated by the testing examples in two existing workflows[3,8], demonstrating comparable localization accuracy while achieving a tenfold improvement in speed. Throughout the experiments, LIRA successfully executed 11 localization steps using fiducial markers and 13 inspection and reasoning tasks, yet still reduced arm manipulation time by 34% compared to the previous solid-state workflow[8].

It is important to clarify that the example shown in Fig. 5 illustrates a representative workflow where LIRA is deployed, rather than a fully autonomous, closed-loop experimental cycle. The goal of this work is to propose a generalizable software framework for vision-based inspection and reasoning in SDLs, with the chosen workflow serving as a demonstrative use case. In a future closed-loop autonomous lab, LIRA would serve as the key perception and reasoning module, continuously analyzing visual input to identify manipulation errors, determine their recoverability, and suggest corrective actions. These structured outputs can feed directly into higher-level workflow engines or schedulers, enabling automated retrying, task reordering, or experiment replanning. While we have demonstrated LIRA's effectiveness in individual robotic tasks-such as object localization, transfer, and error inspection-integrating, the full pipeline from reaction execution to PXRD analysis remains to be demonstrated in future work. The modular design and ROS-based architecture of LIRA ensures compatibility with emerging SDL infrastructure, positioning it as a bridge between real-time perception and autonomous decision-making.

Beyond improving efficiency, LIRA introduces error inspection and reasoning capabilities. The inspection testing in the lab achieved 97.9% success accuracy for the tasks in both workflows[3,8] under naturally varying lighting conditions, confirming the reliability of vision-based inspection. While the current implementation focuses primarily on object placement verification and reasoning, its application can be further extended to additional inspection tasks, such as vial placement and decapping inspection in the solid-state workflow[8].

For the error inspection testing, we observed occasional failures in inspection tasks due to visual ambiguities in specific experimental setups. In particular, LCMS rack misclassification occurred because the sample rack color closely matched the holder's, making it challenging for the model to distinguish between them. This could be mitigated by a simple redesign of rack color in the future. Similarly, errors in NMR rack placement detection were primarily caused by a complex and cluttered background. These challenges highlight the limitations of the current VLM-based approach when applied to visually ambiguous environments.

Future enhancements could expand LIRA's capabilities beyond localized vision-based manipulation, incorporating robot navigation inspection and global path planning using overhead camera systems. By integrating multi-perspective vision inputs, robots could optimize motion planning and collision avoidance, further improving workflow automation. Additionally, the integration of multi-agent collaboration-where multiple robotic systems interact through a shared AI-driven reasoning framework-could enable fully autonomous, self-correcting laboratory environments, pushing SDLs towards greater autonomy and efficiency. For enhancing LIRA's error inspection capabilities, the VLMs such as PaLI-Gemma 2[43], which emphasize high-accuracy visual grounding-could be integrated to improve detection robustness in complex lab environments. Moreover, LIRA's error recovery function relies currently on hardcoded logic to process VLM responses. An end-to-end solution using embodied AI techniques which can directly map from the images to robot actions, such as the RT series models[18,35,36], could further enhance adaptability and autonomy. The current system uses lightweight Gaussian filtering; future iterations could incorporate adaptive or deep-learning-based denoising methods to better handle visually noisy or ambiguous laboratory conditions.

Beyond robotic manipulation, LIRA provides a foundation for developing a centralized AI hub for SDLs, extending beyond inspection and reasoning to full-stack material discovery-from design to synthesis, leveraging both robotics and AI. LIRA is open-source, which ensures that future improvements can be easily iterated and extended, fostering community-driven development of autonomous SDL workflows. The system's Docker-based deployment further simplifies setup and reproducibility, making it accessible for broader adoption in research and industry.

This study establishes a scalable, adaptable foundation for vision-based SDL automation, with broad applications across chemistry and materials discovery, synthesis automation, and real-time experimental adaptation. The continued advancement of AI-driven robotic workflows will further accelerate self-driving lab development, paving the way for fully autonomous, AI-powered scientific discovery. While the current implementation focuses on manipulation and inspection within representative workflows, LIRA has been designed with future end-to-end integration in mind. Specifically, it operates within a ROS-based environment and supports a client-server architecture, allowing seamless communication with external instruments and task schedulers. This makes it straightforward to interface with platforms such as Chemspeed synthesizers and PXRD instruments through a centralized scheduler, enabling coordinated robotic control and real-time data exchange. Full integration into a closed-loop experimental system is a logical next step and part of our ongoing work.

## Methods

### VLM fine-tuning setup

We adopt LLaVA-OneVision 0.5B[41], a lightweight vision-language model that combines a vision encoder with a Qwen2-based language decoder. The model is designed for tasks that require multimodal understanding, making it suitable for error inspection and reasoning in SDLs. The VLM takes as input a camera-captured image and a natural language prompt (e.g., "Has the black rack been placed correctly?") and returns an interpretable sentence describing the placement result and potential recovery options. This approach allows semantic understanding of lab scenes and makes the system generalizable across different error types.

To prepare the model for SDL tasks, we constructed a domain-specific dataset with 540 images labeled with structured prompts and corresponding True/False + reasoning responses. The prompt formatting and annotation procedure is described in Supplementary 5.4. We manually split the data to ensure a balanced distribution across sample rack types, error directions, and locations.

To optimize model efficiency and reduce GPU memory consumption, LLaVA-OneVision 0.5B was fine-tuned using LoRA, applied to both the vision encoder and projector. The LoRA rank was set to 8, with LoRA alpha = 8. The training process used Deepspeed Stage 3 (zero3) optimization, with gradient accumulation steps set to 4. The vision encoder and vision projector were trained, ensuring the model effectively adapted to SDL tasks. Full fine-tuning of the LLM component was performed without Q-LoRA. The model was trained for 12 epochs using a learning rate of $2e-5$ with LoRA dropout set to 0.1, on an NVIDIA GeForce RTX 3080 Ti Laptop GPU (16GB VRAM), taking a total of 115 minutes. Final training and evaluation losses were 0.0123 and 0.0424, respectively, indicating good convergence. Evaluation results are in Supplementary S.5.

### Experiment setup

The LIRA module integrates a camera with an edge-computation device to enable real-time visual localization, inspection, and reasoning. Specifically, the hardware consists of an Intel RealSense D435i RGB-D camera connected via USB 3.0 to an NVIDIA Jetson AGX Orin (64GB RAM), which serves as the computational backbone. Running on Ubuntu 20.04, the Jetson AGX Orin processes visual data for the localization to ensure precise robotic manipulation while also executing the VLM for error detection and reasoning within the LIRA module. The Jetson AGX Orin communicates with the KUKA KMR iiwa robot via a direct LAN connection, ensuring reliable and low-latency control integration. Simultaneously, it connects to other laboratory systems (e.g., LCMS and NMR instruments) over a local 5 GHz Wi-Fi network. In this configuration, cameras mounted on remote instruments stream images wirelessly to LIRA for inspection. LIRA processes the data locally and returns structured reasoning results to remote clients, enabling flexible and responsive remote visual inspection as shown in Fig. 6. This architecture allows seamless data processing and real-time feedback, ensuring robust vision-based task execution across various SDL workflows.

The software framework of the LIRA module is designed for seamless integration and reliable operation in SDLs. Communication between the mobile manipulator and the onboard LIRA module is established via the SOAP protocol, ensuring compatibility with the robot's Java-based applications. For remote interactions with other systems in different workflows, a

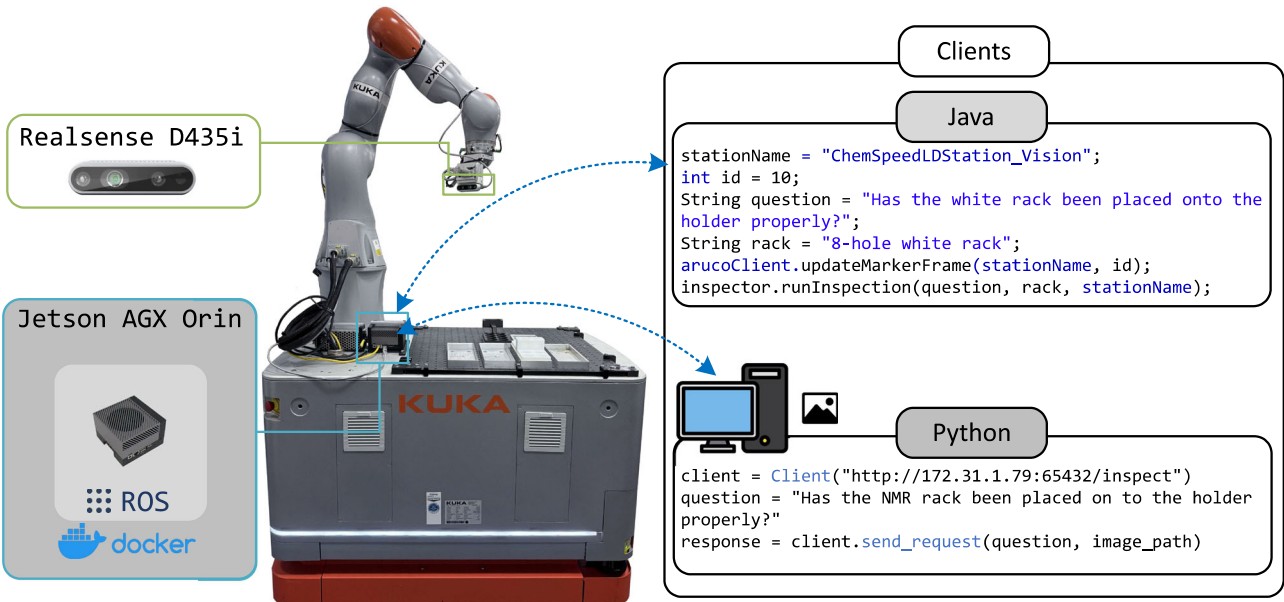

**Fig. 6 |** Mobile manipulator and subsystems.

Flask-based interface is employed, providing a lightweight and efficient HTTP-based communication mechanism. All devices, including the mobile manipulator, LIRA module, and remote systems, operate within a unified network with static IP addresses, simplifying device discovery and ensuring stable, consistent communication. The LIRA module is deployed within a Docker container to improve package management and reproducibility, creating an isolated and portable environment that streamlines installation and compatibility across various platforms. This containerized approach further enhances the system's open-source usability, allowing researchers to replicate and integrate the setup into their own workflows easily.

**Communication interface: client-server framework**

The LIRA module operates within a client-server architecture, supporting both Python and Java clients to enable flexible communication with remote systems. This framework facilitates seamless interaction between the mobile manipulator, the edge computing module, and remote workflows, allowing for real-time inspection and reasoning tasks.

Two client implementations are provided to accommodate different robotic systems and external workflows. The Java client is designed for compatibility with KUKA Sunrise applications, utilizing the SOAP protocol to enable direct communication with the Java-based framework of the KUKA manipulator. This client captures an image, queries the LIRA server, and receives an inspection response. The processing is handled by the InspectionHandler module, which determines whether the workflow should proceed or initiate corrective actions. An example Java client request is shown in Fig. 6a:

```
// Java Client Example
stationName = "ChemSpeedLDStation_Vision";
int id = 10;
String question = "Has the 8-hole white rack been placed onto the holder properly?";
String target = "white_rack";
arucoClient.updateMarkerFrame(stationName, id);
inspector.runInspection(question, target, stationName, id);
```

For external systems and flexible deployment, a Python-based client communicates with LIRA through a lightweight Flask-based HTTP interface. The Python client captures an image using a local camera and transmits

an inspection prompt as a natural language query. The server processes the request and returns a binary classification result (True or False), indicating whether the manipulation task was executed correctly. If an error is detected, the reasoning module provides additional contextual information on the misalignment and potential corrective actions. An example Python client request is shown in Fig. 6:

```
# Python Client Example
client    =    Client("http://172.31.1.79:65432/inspect")
question = "Has the NMR/white long rack been placed onto the holder properly?"
response    =    client.send_request(question, image_path)
```

The server-side processing is handled by a Flask-based web service running on the Jetson AGX Orin, which executes image-based inspection and reasoning tasks. Upon receiving a request, the server extracts the prompt and image, processes them using the VLM, and returns a response to the client, indicating whether the manipulation was successfully executed or required correction. The server integrates with a ROS-based image acquisition pipeline for Java-based SOAP requests, capturing an image upon request, analyzing it, and sending back the inspection results.

A typical workflow execution begins with the robot performing a manipulation task, such as placing a rack or transferring an object. The inspection client then captures an image and sends a query to LIRA, such as "Has the white 8-hole rack been placed onto the holder properly?". The LIRA module processes the request, applies VLM-based reasoning, and returns an inspection result along with error reasoning if misalignment is detected. Based on this feedback, the robot follows a predefined recovery strategy to correct any placement errors before resuming the workflow which is detailed in Supplementary S.7.

This client-server interface enables remote monitoring, multi-platform integration, and real-time decision-making, significantly improving workflow automation in SDL environments. LIRA offers a flexible, extensible solution for integrating AI-driven inspection and reasoning into robotic workflows by supporting SOAP-based Java and Flask-based Python communication.

To evaluate communication performance, we measured the transmission latency of both client types under realistic conditions. The Java

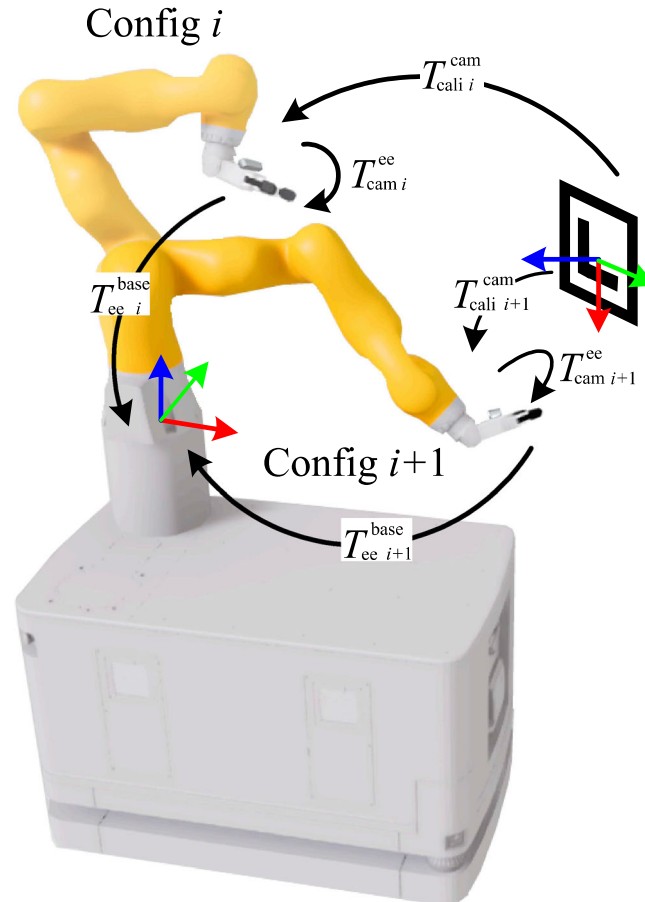

## Config *i*

$T_{\text{cali }i}^{\text{cam}}$

$T_{\text{cam }i}^{\text{ee}}$

$T_{\text{ee }i}^{\text{base}}$

$T_{\text{cali }i+1}^{\text{cam}}$

$T_{\text{cam }i+1}^{\text{ee}}$

## Config *i+1*

$T_{\text{ee }i+1}^{\text{base}}$

**Fig. 7 | Schematic diagram of calculating transformation from the tool frame of the robot to the camera frame.**

client communicates via SOAP over a wired Ethernet connection to the Jetson server, achieving average latencies of ~28 ms for image transmission and 12 ms for text-based inspection queries. In contrast, the Python client communicates via Flask over a 5G router on a local wireless network, yielding latencies of ~45 ms for images and 15 ms for text. These results confirm that both implementations are suitable for real-time inspection tasks in SDL workflows, with acceptable latency under typical lab configurations.

### Detection preprocessing for fiducial marker-based localization

To ensure accurate and stable marker detection, a two-step filtering approach is applied, consisting of Gaussian filtering for noise reduction and a buffered detection refinement process to enhance pose estimation. This method mitigates the effects of motion blur, lighting variations, and transient detection errors, improving visual localization precision for robotic manipulation.

In the image preprocessing stage, the input image undergoes Gaussian filtering with a 5 × 5 kernel and zero standard deviation, implemented using OpenCV's `cv2.GaussianBlur()` function. Gaussian filtering is applied to smooth spatial noise in the pixel domain by weighting neighboring pixels with a Gaussian kernel. This helps reduce high-frequency visual artifacts such as lighting flicker and marker edge jitter, which can degrade pose estimation accuracy. After filtering, the image is processed using OpenCV's ArUco detection pipeline, where a predefined dictionary (`DICT_6X6_50`) is loaded for marker recognition. The detection parameters include subpixel corner refinement (`CORNER_REFINE_SUBPIX`) to improve localization accuracy, followed by pose estimation using `cv2.aruco.estimatePoseSingleMarkers()`. The rotation

vector (`rvec`) obtained from this function is converted into a quaternion representation for further processing.

Once a marker is detected, an additional filtering step is applied to improve localization stability. This process, handled by the `filter_detections()` function, maintains a buffer of recent detections for each marker. The mean pose is computed by averaging the position and orientation values of all buffered detections. To further refine the pose estimation, the detection closest to the computed mean pose is selected by minimizing the Euclidean distance between each buffered detection and the averaged pose. The function returns the closest detection, reducing transient errors and ensuring smooth pose estimation.

By combining Gaussian filtering for noise reduction with buffered detection refinement, the proposed preprocessing method enhances the precision and robustness of vision-based localization for robotic systems. This approach is particularly effective in dynamic SDL environments where stable and reliable fiducial marker detection is crucial for seamless robotic manipulation and object interaction.

### Transformation from the tool frame of the robot to the camera frame

To map ArUco marker frames to the robot's coordinate system, the camera-to-end-effector transformation matrix ($T_{\text{cam}}^{\text{ee}}$) must first be determined. This is achieved via eye-in-hand calibration, where the robot captures images of a stationary calibration board from multiple angles as shown in Fig. 7. For each pose *i*, we record: 1) End-effector pose relative to the robot base $T_{\text{ee}}^{\text{base}}$ *i*. 2) Calibration board pose in the camera frame $T_{\text{cali}}^{\text{cam}}$ *i*.

Solving the least squares formulation from multiple poses yields the optimal transformation matrix $T_{\text{cam}}^{\text{ee}}$, enabling precise conversion of object poses into the robot's base frame for manipulation:

$$T_{\text{targets}}^{\text{base}} = T_{\text{ee}}^{\text{base}} \cdot T_{\text{cam}}^{\text{ee}} \cdot T_{\text{cali}}^{\text{cam}} \cdot T_{\text{targets}}^{\text{cali}} \tag{1}$$

Tactile-based calibration[3,7,8] updates $T_{\text{marker}}^{\text{ee}}$ via force-based cube interactions, requiring sequential movements and sensor validation, making it slower. In contrast, vision-based calibration using ArUco markers is faster and more flexible but relies on visual data quality, which can be affected by lighting, reflections, and computational load.

### Data availability
The data and codes that support the findings of this study are available in the repository1 with the identifier https://doi.org/10.5281/zenodo.17108096.

### Code availability
The code supporting this study is openly available at the link1. Users can reproduce the inference testing and evaluation described in the manuscript by following the instructions and dataset structure provided in the repository. Details on software versions, dependencies, and parameter settings are included within the repository documentation.

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

## Acknowledgements

We acknowledge financial support from the Leverhulme Trust via the Leverhulme Research Centre for Functional Materials Design. The authors also received funding from the Engineering and Physical Sciences Research Council (EPSRC, EP/V026887/1). AIC thanks the Royal Society for a Research Professorship (RSRP\S2\232003).

## Author contributions

A.I.C. supervised and directed the research. Z.Z. proposed LIRA, developed the software, trained VLM, collected image data, wrote the draft and tested the mobile robot platform. S.V. tested the vision-based localization method to compare with tactile-based method and collected image data for NMR and LCMS inspection. F.M. designed the mounting frame for the Jetson and configured its power connection to the mobile robot. H.F. supervised the software development of LIRA and hardware testing for the SDL application. All the authors provided feedback on both the research and the draft.

## Competing interests

The authors declare no competing interests.
