## [Transparent Peer Review file · Communications Chemistry]

This manuscript has been previously reviewed at another journal. This document only contains information relating to versions considered at Communications Chemistry.

LIRA: Localization, Inspection, and Reasoning Module for Autonomous Workflows in Self-Driving Laboratories

Corresponding Author: Professor Andrew Ian Cooper

Version 0:

Reviewer comments:

Reviewer #1

(Remarks to the Author)

This paper presents a LIRA module that integrates visual localization, inspection, and reasoning to enhance workflows in Self-Driving Labs (SDLs), which is a forward-looking research direction. However, the current version of the manuscript suffers from a lack of detail regarding key methods, the experimental setup, and the specific operational workflows. This deficiency undermines the scientific rigor and reproducibility of the work. The authors need to supplement several core sections with crucial technical details and evaluation procedures to fully demonstrate the innovation and effectiveness of their approach. To meet the standards for publication, it is recommended that the authors undertake a major revision based on the specific points outlined below.

1.Regarding Localization Precision Evaluation (Section 2.2): The manuscript's description of the localization precision evaluation methodology is insufficiently detailed. The authors should elaborate on the entire evaluation process to enhance the persuasiveness of their conclusions. Specifically, the following should be detailed: How the reference position was defined (e.g., 300 mm away from the calibration position) and how the ground truth was obtained. The results of tactile calibration should also be used as reference data.

2.Regarding the VLM Fine-Tuning Process (Section 2.4): The description of the fine-tuning process for the Vision Language Model (VLM) used for inspection and reasoning is inadequate. The authors need to provide more details about the dataset construction and model training. It is recommended to add the following information:

A detailed explanation of how the fine-tuning dataset was collected, including its contents (e.g., properly placed and misaligned sample racks), image resolution (640x480 pixels).

Figure 4.a shows "LRBR". The author needs to confirm whether "F" is missing.

A detailed description of the structured data annotation process and the prompt design used for fine-tuning the VLM inspector. A concrete example of a standardized human prompt should be provided so that readers can understand how accurate error detection, reasoning, and recovery are achieved.

3.Regarding the Specific Role of LIRA in the Workflow: The manuscript fails to clearly articulate the precise role of the LIRA module and its closed-loop verification mechanism within the overall operational workflow. The authors should detail: How LIRA's inspection and reasoning functions verify each placement manipulation in a closed-loop manner. The process by which the robotic arm moves to a predefined position to perform corrective actions based on the generated prompts, and how this process is repeated iteratively until LIRA returns a "True" result.

4.Regarding the Experimental Setup (Section 4.2): The description of the experimental platform is too brief and lacks critical hardware and architectural information, which impacts the work's reproducibility. The authors should provide a detailed description of the setup, including:

The specific hardware integrated into the LIRA module.

An explanation of how this architecture enables seamless data processing and real-time feedback to ensure robust vision-based task execution across various SDL workflows.

Elaboration on how this edge-computing solution communicates wirelessly with remote clients to support remote visual inspection.

5.Regarding Figure Descriptions (Figure 5): As a core illustration showing the LIRA-enhanced SDL workflow, the description for Figure 5 should be more comprehensive. It is recommended that the authors explain more clearly in the main text how each sub-figure (a, b, c) specifically demonstrates LIRA's application in localization, inspection, and reasoning. As shown in Figure 5, positioning and detection do not use the same camera. Where are they installed and what tasks do they perform? Is there a separation of positioning, detection, and inference functions?

The authors' response to Reviewer 3 provides a partial explanation for the lack of an end-to-end automation implementation. However, the following issues should still be carefully considered:

1. Although the authors have clarified that the LIRA module has not yet been integrated into a fully end-to-end system, a more explicit and detailed description is needed regarding the role this module plays within a closed-loop autonomous laboratory system. Furthermore, the authors should elaborate on how LIRA could be incorporated into a future fully automated assistant system and clearly demonstrate its significance.

2. While the individual operational verifications can validate the functionality and a certain degree of stability in tasks such as localization, perception, and reasoning, how does the system perform in continuous tasks? For example, how well can localization and perception be integrated to execute a complete operation?

3. In the context of an autonomous laboratory workflow, modular programming is a feasible and common approach for the progressive development of automation systems. However, in order to align with the broader vision proposed in Professor Cooper's Nature paper, how can LIRA be directly integrated into the existing system and fulfill its intended functionalities?

4. To appeal to researchers from diverse backgrounds, especially those in the field of chemistry, the manuscript should place greater emphasis on the description of chemical scenarios and clearly articulate their significance.

Version 1:

Reviewer comments:

Reviewer #1

(Remarks to the Author)

The questions raised have been answered and resolved, and I agree to accept.

Response to Reviewers:

LIRA: Localization, Inspection, and Reasoning Module for Autonomous Workflows in Self-Driving Labs

Note: Newly added content in response to the reviewer comments is highlighted in the manuscript in blue, and deleted content has been highlighted in red to identify the changes rapidly. Other changes in the manuscript are marked with a numbered blue balloon, like so ①, for easy reference. The marked-up manuscript is appended at the end of this response. All references in this response refer to the revised manuscript (the numbering has changed).

Comments by Reviewer #1

This paper presents a LIRA module that integrates visual localization, inspection, and reasoning to enhance workflows in Self-Driving Labs (SDLs), which is a forward-looking research direction. However, the current version of the manuscript suffers from a lack of detail regarding key methods, the experimental setup, and the specific operational workflows. This deficiency undermines the scientific rigor and reproducibility of the work. The authors need to supplement several core sections with crucial technical details and evaluation procedures to fully demonstrate the innovation and effectiveness of their approach. To meet the standards for publication, it is recommended that the authors undertake a major revision based on the specific points outlined below.

1.Regarding Localization Precision Evaluation (Section 2.2): The manuscript's description of the localization precision evaluation methodology is insufficiently detailed. The authors should elaborate on the entire evaluation process to enhance the persuasiveness of their conclusions. Specifically, the following should be detailed: How the reference position was defined (e.g., 300 mm away from the calibration position) and how the ground truth was obtained. The results of tactile calibration should also be used as reference data.

We thank the reviewer for highlighting this point. To address your concern comprehensively, we have provided additional methodological details in Section 2.2 of the revised manuscript, explicitly marked at position ① on page 4. Specifically, we clarified how the 300 mm reference position was determined using the robot's internal positioning capabilities, explicitly referencing the documented ± 0.15 mm pose repeatability (ISO 9283) of our KUKA IIR iiwa 14 robotic arm. Additionally, we detailed the procedure used to establish the ground truth via tactile calibration employing a calibration cube, which has been extensively validated previously ([7,8]). The tactile calibration results were explicitly used as reference data for benchmarking our vision-based calibration method. We believe these revisions substantially enhance the clarity and persuasiveness of our evaluation methodology.

2.Regarding the VLM Fine-Tuning Process (Section 2.4): The description of the fine-tuning process for the Vision Language Model (VLM) used for inspection and reasoning is inadequate. The authors need to provide more details about the dataset construction and model training. It is recommended to add the following information:

A detailed explanation of how the fine-tuning dataset was collected, including its contents (e.g., properly placed and misaligned sample racks), image resolution (640x480 pixels).

Figure 4.a shows "LRBR". The author needs to confirm whether "F" is missing.

A detailed description of the structured data annotation process and the prompt design used for fine-tuning the VLM inspector. A concrete example of a standardized human prompt should be provided so that readers can understand how accurate error detection, reasoning, and recovery are achieved.

We thank the reviewer for the constructive suggestions. We have substantially revised Section 2.4 to address all requested points. Specifically, at ② on page 6, we now describe in detail how the dataset was constructed, including the use of a camera mounted on the robot's end effector, image resolution (640x480 pixels), and the inclusion of both properly placed and misaligned sample racks across multiple types (8-hole racks, PXRD plates, NMR racks, and LCMS racks). We also clarify that misalignments were systematically introduced along four

translational directions (Left, Right, Forward, Backward) and rotational deviations. Regarding Figure 4.a, we confirm that the omission of “F” was unintentional, and we have corrected the figure to include all directions as “LRFB+R.”

At ③ on page 7, we now explain the structured data annotation process in detail. Each image was manually labeled by an experienced operator to indicate (1) placement correctness, (2) misalignment direction, and (3) recoverability status. These labels were then paired with standardized natural language prompts to supervise fine-tuning. We also provide a concrete example of a prompt and its corresponding structured responses (True/False with reasoning and recoverability assessment), illustrating how the VLM achieves accurate error detection and decision-making.

We believe these additions address the reviewer’s concerns and significantly improve the clarity and completeness of the VLM fine-tuning description.

3. Regarding the Specific Role of LIRA in the Workflow: The manuscript fails to clearly articulate the precise role of the LIRA module and its closed-loop verification mechanism within the overall operational workflow. The authors should detail:

-How LIRA's inspection and reasoning functions verify each placement manipulation in a closed-loop manner.

-The process by which the robotic arm moves to a predefined position to perform corrective actions based on the generated prompts, and how this process is repeated iteratively until LIRA returns a "True" result.

We appreciate the reviewer’s comment. To clarify the role of LIRA in the workflow, we have revised Section 2.5 and added a new paragraph at ④ on page 9. The updated text now explains in detail how LIRA performs closed-loop verification for each object placement. Specifically, after each placement, the robot moves to a predefined inspection pose, captures an image, and sends it to LIRA. LIRA responds with a structured output indicating placement correctness, misalignment direction, and recoverability status (as shown in Figure 4b). If the response is "False" but recoverable, the robot performs a corrective motion based on the reasoning output, and this inspection–correction loop continues until a "True" result is returned. If the error is unrecoverable, the system pauses for manual intervention. This closed-loop mechanism ensures robust and autonomous correction of placement errors, thereby enhancing the reliability of the overall workflow. We believe this addition addresses the reviewer’s concern and improves the clarity of LIRA’s operational role.

4. Regarding the Experimental Setup (Section 4.2): The description of the experimental platform is too brief and lacks critical hardware and architectural information, which impacts the work's reproducibility. The authors should provide a detailed description of the setup, including:

-The specific hardware integrated into the LIRA module.

-An explanation of how this architecture enables seamless data processing and real-time feedback to ensure robust vision-based task execution across various SDL workflows.

-Elaboration on how this edge-computing solution communicates wirelessly with remote clients to support remote visual inspection.

We thank the reviewer for the constructive feedback. In Section 4.2, we have significantly expanded the description of our experimental setup. As detailed at ⑤ on page 12, we now specify the hardware used in the LIRA module (Intel RealSense D435i camera and Jetson AGX Orin 64GB), the LAN and 5 GHz Wi-Fi network topology, and the data flow between the robot, edge processor, and remote systems. We also describe the communication architecture, including SOAP for robot control and a Flask-based HTTP API for remote clients. These updates clarify how real-time vision-based processing and remote visual inspection are achieved through our edge-computing architecture and ensure that the system can be reliably reproduced by other researchers.

5. Regarding Figure Descriptions (Figure 5): As a core illustration showing the LIRA-enhanced SDL

workflow, the description for Figure 5 should be more comprehensive. It is recommended that the authors explain more clearly in the main text how each sub-figure (a, b, c) specifically demonstrates LIRA's application in localization, inspection, and reasoning. As shown in Figure 5, positioning and detection do not use the same camera. Where are they installed and what tasks do they perform? Is there a separation of positioning, detection, and inference functions?

Thank you for this helpful suggestion. We have revised the beginning of Section 3.2 (now marked as ⑥ on page 7) to provide a clearer and more structured explanation of each sub-figure in Figure 5. Specifically, we now explicitly describe how:

- Figure 5a illustrates the full pipeline of the solid-state workflow, emphasizing LIRA's role in fiducial-marker-based localization;
- Figure 5b highlights inspection and reasoning using images captured by a camera mounted on the robot's end effector;
- Figure 5c presents a remote inspection scenario where fixed cameras at LCMS and NMR stations stream images to LIRA for reasoning.

To address the question about camera placement and functional separation, we have added detailed clarification on the camera configurations used across workflows. As described in ⑥, a RealSense D435i camera is mounted on the robot's end effector for close-range post-placement inspection and localization, while fixed webcams are installed above remote instruments for distributed, remote inspection. All image data—regardless of source—is processed by a unified LIRA module on the Jetson AGX Orin, ensuring consistent and centralized reasoning logic across embodiments.

We believe these additions directly address the reviewer's concerns and enhance the clarity of Figure 5 and its role in illustrating LIRA's multi-context application.

The authors' response to Reviewer 3 provides a partial explanation for the lack of an end-to-end automation implementation. However, the following issues should still be carefully considered:

1. Although the authors have clarified that the LIRA module has not yet been integrated into a fully end-to-end system, a more explicit and detailed description is needed regarding the role this module plays within a closed-loop autonomous laboratory system. Furthermore, the authors should elaborate on how LIRA could be incorporated into a future fully automated assistant system and clearly demonstrate its significance.

We appreciate the reviewer's suggestion and have revised both the Introduction and Discussion sections to more clearly articulate the envisioned role of LIRA in a closed-loop autonomous laboratory system.

- In the Introduction (see ⑦ on page 2), we now explain that LIRA is designed not only for real-time inspection and reasoning but also to serve as a modular reasoning module capable of bridging perception and decision-making. We emphasize that its structured outputs—such as error classification and recovery strategy—can interface directly with high-level planners or workflow engines, enabling autonomous replanning and error correction.
- In the Discussion section (see ⑧ on page 10), we further elaborate that LIRA's structured reasoning results could drive automated task retries, dynamic task reordering, or experiment replanning in future iterations. We highlight its ROS-based architecture and modular design, which position it well for seamless integration with emerging SDL infrastructures as they evolve toward fully autonomous operation.

These additions aim to make LIRA's significance in a future end-to-end SDL system more concrete and demonstrate its value as a foundational module for autonomous decision-making in robotic workflows.

2. While the individual operational verifications can validate the functionality and a certain degree of

stability in tasks such as localization, perception, and reasoning, how does the system perform in continuous tasks? For example, how well can localization and perception be integrated to execute a complete operation?

We appreciate the reviewer's interest in assessing the system under continuous operational conditions. To clarify, all validation experiments described in the manuscript were conducted within real-world, integrated SDL workflows. Both localization and inspection were evaluated not as isolated functions but as embedded components actively triggered during end-to-end robotic task execution.

The LIRA module was invoked directly from the robot-side programs through the inspection handler interface (see Supplementary Section A.7), enabling tight integration into the robot control flow. This architecture allowed seamless coordination between the robot and the perception-reasoning pipeline. Our 200 test cases across both workflows were performed using this setup, simulating realistic failure modes by introducing deliberate placement errors and evaluating LIRA's ability to detect and reason about them in-line with the robot's actions.

In addition, our runtime benchmarking compared the complete solid-state workflow before and after LIRA integration, covering all localization, perception, and reasoning steps executed continuously during real robotic manipulation. These results demonstrate not only the effectiveness of each module but also their reliability under system-level, closed-loop operation.

We have updated the manuscript accordingly (see ⑨ on page 10) to clearly state that all tests reflect integrated, continuous workflow execution rather than standalone module evaluation.

3. In the context of an autonomous laboratory workflow, modular programming is a feasible and common approach for the progressive development of automation systems. However, in order to align with the broader vision proposed in Professor Cooper's Nature paper, how can LIRA be directly integrated into the existing system and fulfill its intended functionalities?

We appreciate the reviewer's thoughtful comment regarding LIRA's integration within a broader autonomous laboratory architecture, particularly in alignment with the vision presented in Professor Cooper's *Nature* paper.

To clarify, LIRA was explicitly designed with modularity and system compatibility in mind. Both the localization and inspection functionalities are implemented as ROS-based service clients, following the same modular programming approach used in our previous tactile-based calibration pipeline. In the KUKA control environment, these clients are invoked through package imports within the Java-based robot program—ensuring minimal changes to the existing task structure when upgrading from tactile to vision-based methods.

In the context of the *Nature* workflow, integrating LIRA would involve inserting a lightweight inspection node at the appropriate step in the task sequence (e.g., post-placement or pre-measurement) without altering the upstream or downstream logic. This is precisely how LIRA was deployed in our experimental workflows: we selected workflows already in use in our laboratory and integrated LIRA into them with minimal modification to their overall structure. This not only demonstrates LIRA's compatibility with existing autonomous systems but also validates its robustness in real-world, continuous operation.

We have revised the manuscript (see ⑨ on page 10 and Supplementary A.7) to better highlight how LIRA is designed to plug into existing workflows as a modular, vision-driven enhancement without requiring complete workflow redesign.

4. To appeal to researchers from diverse backgrounds, especially those in the field of chemistry, the manuscript should place greater emphasis on the description of chemical scenarios and clearly articulate their significance.

We thank the reviewer for this valuable suggestion. To better emphasize the chemical relevance of our work, we have revised both the introduction and the experimental validation section to more clearly describe the real-world chemical contexts where LIRA operates.

Specifically, in the introduction (see ⑩ on page 1), we now highlight that workflows involving PXRD plate insertion and LCMS/NMR vial analysis are particularly sensitive to manipulation errors, which can lead to failed measurements or even equipment damage. We clarify that these scenarios demand robust error inspection systems, thus underscoring the significance of LIRA's application in chemistry labs.

In the evaluation section (Figure 5, see ⑪ on page 7), we describe how LIRA supports two representative chemistry-focused workflows: (1) a solid-state materials workflow where it ensures correct PXRD plate placement, and (2) a synthesis workflow where it validates vial rack positioning before LCMS and NMR analysis. These examples help demonstrate the practical importance of LIRA for safe and accurate operation in real chemical experiments.

We hope these clarifications will better contextualize LIRA's utility and importance for researchers in chemistry and materials science.